# InGaN Nanorods Decorated with Au Nanoparticles for Enhanced Water Splitting Based on Surface Plasmon Resonance Effects

**DOI:** 10.3390/nano10050912

**Published:** 2020-05-09

**Authors:** Qing Liu, Jiang Shi, Zhenzhu Xu, Bolin Zhang, Hongliang Liu, Yinlei Lin, Fangliang Gao, Shuti Li, Guoqiang Li

**Affiliations:** 1Guangdong Engineering Research Center of Optoelectronic Functional Materials and Devices, Institute of Semiconductors, South China Normal University, Guangzhou 510631, China; 2018022795@m.scnu.edu.cn (Q.L.); 2018022807@m.scnu.edu.cn (J.S.); berlinzhang@foxmail.com (B.Z.); 2019022815@m.scnu.edu.cn (H.L.); lishuti@scnu.edu.cn (S.L.); 2State Key Laboratory of Luminescent Materials and Devices, South China University of Technology, Guangzhou 510640, China; zhenzhupearl@163.com; 3School of Materials Science and Energy Engineering, Foshan University, Foshan 528000, China; linyinlei@fosu.edu.cn

**Keywords:** PEC water splitting, InGaN nanorods, Au nanoparticles, surface plasmon resonance

## Abstract

Photoelectrochemical (PEC) water splitting has great application potential in converting solar energy into hydrogen energy. However, what stands in the way of the practical application of this technology is the low conversion efficiency, which can be promoted by optimizing the material structure and device design for surface functionalization. In this work, we deposited gold nanoparticles (Au NPs) with different loading densities on the surface of InGaN nanorod (NR) arrays through a chemical solvent route to obtain a composite PEC water splitting system. Enhanced photocatalytic activity, which can be demonstrated by the surface plasmon resonance (SPR) effect induced by Au NPs, occurred and was further confirmed to be associated with the different loading densities of Au NPs. These discoveries use solar water splitting as a platform and provide ideas for exploring the mechanism of SPR enhancement.

## 1. Introduction

The efficient capture and conversion of solar energy with semiconductors has drawn tremendous interest, for it generates clean and sustainable hydrogen energy without consuming electricity [1,2]. In this regard, solar water splitting provides a feasible way to achieve this vision [3]. Photoelectrochemical (PEC) water splitting converts solar energy through three basic processes: (1) light absorption and excitation of photogenerated carriers, (2) separation of photogenerated carriers and migration to the semiconductor surface, and (3) redox reactions induced by photogenerated carriers [1]. Since Fujishima and Honda used TiO_2_ to achieve PEC water splitting in 1972, researchers have made unremitting efforts to develop high-efficiency solar-hydrogen conversion materials [4].

Among numerous semiconductor materials, III-nitride semiconductors have been regarded as promising candidates for PEC water splitting; this originates from their tunable energy bandgaps, which span nearly the entire solar spectrum [5,6,7]. In addition, the band edge has good energy alignment with the water redox potential, which meets the thermodynamic requirements of water splitting [1,7,8]. So far, research based on III-nitride semiconductors has been carried out through various low-dimensional structures. Compared with their bulk and film, low-dimensional structures have a larger active area and lower light reflectivity, which can strengthen light absorption [3,9,10]. Surface band bending is also beneficial for improving carrier separation efficiency and reducing the recombination probability during migration [11,12]. Furthermore, the transmission distance of photogenerated carriers to the semiconductor–electrolyte interface will be greatly reduced with a lower-dimensional structure [13,14].

Functionalizing semiconductor surfaces with plasmonic nanoparticles (NPs), which might cause surface plasmon resonance (SPR) after irradiation, can further enhance light absorption and carrier density [15,16,17]. The combination of semiconductors and plasmonic NPs constructs a composite material system in which plasmon resonance of NPs enhances the PEC performance based on four significant mechanisms: (1) hot electron injection (HEI) enhancement induced by SPR, (2) light-scattering (trapping) enhancement, (3) plasmon resonance energy transfer (PRET), and (4) light-concentrating enhancement [18,19,20]. The essence of HEI enhancement is that NPs are excited when electrons are transferred to the conduction band of the semiconductor. In this mechanism, photons with energy below the semiconductor bandgap but higher than the metal–semiconductor Schottky barrier can also contribute to PEC water splitting [15]. Light-scattering enhancement occurs because NPs will first scatter incident light onto the semiconductor surface, thereby reducing the reflection of light from the semiconductor. This effect is more prominent with NPs that are larger than 100 nm [21,22]. Plasmon resonance energy transfer enhances the photocatalytic activity of semiconductors by enhancing carrier generation [23]. Finally, the interaction of NPs and light leads to an enhancement of local electromagnetic fields on the semiconductor surface, which in turn results in the light concentration effect.

To date, numerous reports on NP-enhanced water splitting have illustrated the interest in this method. For instance, Kang et al. reported an Au/ZnO composite that exhibits enhanced solar-to-hydrogen efficiency of 0.52%, compared with that of bare ZnO (0.24%); they revealed that the enhancement is caused by light-scattering and HEI [24]. Similarly, Qian et al. reported that the size of Au NPs plays an important role in the decoration of TiO_2_, which is directly related to the strength of the HEI effect [25]. Shin et al. also found that the photocatalytic efficiency of Au NPs that decorated ZnO-TiO_2_ was promoted because of HEI, and that the effect is loading density-dependent [26]. Hou reported that the photo-to-current efficiency of the TiO_2_/Au/InGaN sandwich structure was enhanced, which can be explained by HEI [27]. Claverie et al. observed an HEI effect on Au/TiO_2_, and the absorption can be increased by more than 40 times [28]. Liu et al. reported that Au/Ni NPs deposited on a TiO_2_/n-Si photoanode have 3-fold efficiency under illumination, which can be ascribed to the PRET effect [29]. Basu et al. revealed that ZnO nanosheets decorated with Au NPs have evident light-scattering and light-absorbing effects [15]. Sang et al. found that the incident photon-to-electron conversion efficiency of InGaN/GaN multiple-quantum-well nanorod-based photoelectrodes (PEs) reached up to 64% after coupling with Ag–Au core–shell nanowires [30].

From the published literature, it can be inferred that NPs have a great effect on the improvement of photocatalytic activity, and this effect is closely related to the size and loading density of the NPs. In this work, InGaN nanorod (NR) arrays decorated with Au NPs were fabricated. NR arrays were first synthesized through molecular-beam epitaxy (MBE), and Au NPs of different loading densities were then deposited. The results indicate that InGaN NRs decorated with Au NPs have a better PEC activity compared to bare InGaN NRs, and that the performance can be optimized by using different loading densities. This enhancement is mainly attributed to the SPR effect from Au NPs.

## 2. Experimental Section

### 2.1. MBE Growth of InGaN NRs on Si (111) Wafers

The Si (111) wafers were washed three times with acetone, isopropanol, deionized water, and dried with N_2_ before preparing the InGaN NRs. A radio-frequency plasma-assisted MBE system (PA-MBE, MANTIS, UK) was used to grow InGaN NRs on bare Si wafers. The growth conditions of InGaN NRs were fixed on an in-beam equivalent pressure (BEP) of 4 × 10^−7^ Torr, Ga BEP of 7.2 × 10^−8^ Torr, a forward plasma power of 400 W, and a N_2_ flux of 2.0 sccm. The temperature and time required for growth were 900 °C and 3 h, respectively, as described previously [31,32,33,34,35]. 

### 2.2. Decoration of Au NPs on InGaN NRs

Au NPs were deposited on InGaN NRs with ionic layer adsorption and thermal-reduction methods. First, InGaN NRs were immersed into the HAuCl_4_ aqueous solution for 12 h in the dark and then blown dry with N_2_. Au NPs with different particle sizes and growth densities were obtained by using different concentrations of HAuCl4 aqueous solution. The concentrations of the HAuCl_4_ aqueous solutions were 5, 10, and 15 μM and were marked as Au5@NRs, Au10@NRs, and Au15@NRs, respectively. A mixed solution of 4 mL of absolute ethanol and 6 mL of deionized water was heated using a device (HWJB-2100CT, Zhengzhou Carbon State Instrument Co., Ltd., China) set to 110 ℃. The as-grown wafer was immersed in the mixed solution for 10 min. Finally, the as-grown wafer was washed with deionized water and dried with N_2_.

### 2.3. Materials Characterization

The morphology of InGaN NRs was characterized by scanning electron microscopy (SEM, ZEISS Ultra 55, Oberkochen, Germany) and transmission electron microscopy (TEM JEOL 3000F, FEI, Japan). The UV-vis absorption spectrum of the sample was characterized using an UV-vis spectrometer (Cary 500, Varian, Palo Alto, CA, USA). 

### 2.4. PEC Measurements

PEC measurements were carried out on a three-electrode electrochemical workstation (CHI 760E, CH Instruments Inc., Austin, TX, USA) In the PEC system of this experiment, the as-prepared sample was the working electrode, the platinum wire was the counter electrode, the saturated calomel electrode (saturated KCl, SCE) was the reference electrode, and the electrolyte was a 0.5 M H_2_SO_4_ (pH = 0) solution. A 300 W Xe lamp (100 mW cm^−2^) was used to illuminate the sample. The working area of the photoanode was 0.3–0.6 cm^2^. Unless otherwise specified, all potentials in this experiment were reported with respect to SCE.

## 3. Results and Discussion

The design and fabrication process of decorating Au NPs on InGaN NR arrays is shown in Figure 1. It can be found from the figure that the complete experiment is mainly divided into two parts. First, MBE was used to grow an InGaN NR array on a Si substrate, and then the NRs were decorated with Au NPs using ion exchange. The detailed growth process and steps can be seen in the experimental section. 

The surface morphology of semiconductor materials is very important for the PEC water splitting system. From Figure 2a, it can be obtained that the prepared InGaN NRs have high density. The surface of the InGaN NRs is very smooth, which indicates the high quality of the as-grown InGaN NRs. The Au NPs decorated on the surfaces of the InGaN NRs had different loading densities, and the corresponding SEM images are shown in Figure 2b–d. From the figure, it can be seen that the SEM images of the deposited Au NPs have a clear graininess compared to those of the undeposited, and the graininess becomes stronger as the concentration of the HAuCl_4_ aqueous solution increases. This is mainly attributed to the fact that more Au NPs are deposited on the nanopillars under the same conditions through the ion exchange reaction. However, one cannot determine from the SEM images whether Au NPs are deposited on the InGaN NRs, and the size of Au NPs cannot be observed. Therefore, TEM was used for further characterization. Figure 2e, f shows the High-resolution TEM (HRTEM) diagram of a single Au@NR, in which it can be clearly seen that there is good contact between the Au NPs and the InGaN NRs. Figure 2f shows the well-resolved highly crystalline structure along the (0001) plane of InGaN NRs (d = 0.536 nm), as well as along the (111) plane of the face-centered cubic Au (d = 0.24 nm), and the diameter of the Au NPs is ~5 nm. For samples Au10@NR and Au15@NR, their particle sizes are 11 and 16 nm, respectively. An energy dispersive spectrometer (EDS, ZEISS Ultra 55, Oberkochen, Germany) was used for verification of the densities of the Au NPs. Appendix A shows the EDS diagram of InGaN NRs with Au NPs deposited using HAuCl_4_ aqueous solutions of different concentrations. EDS analysis indicates that the sample has a high Au content, and the Au content increases as the concentration of the immersion solution increases. This is consistent with the previous inference.

The light absorption characteristic of the InGaN NRs photoelectrode is very important for the efficiency of PEC water splitting. Figure 3 shows the absorption spectra of bare NRs, Au5@NRs, Au10@NRs, and Au15@NRs. It can be seen from the figure that InGaN has an absorption center at 460 nm, and an absorption cut-off peak around 650 nm. This absorption spectrum of InGaN NRs is mainly caused by the direct band-to-band transition [31]. However, after the Au NPs are deposited on the InGaN NRs, the intensity of the absorption peak decreases as the deposition density increases. After the NPs are deposited, the absorption of light on the surface of the InGaN NRs is hindered, which affects the direct band-to-band transition and weakens the absorption peak. In addition, the broad long-wavelength absorption is attributed to a spatially indirect transition [31]. Thus, InGaN NRs have better light absorption in the near-infrared short-wave region (780–1100 nm), whereas there is almost no significant absorption in the subsequent regions, which can be clearly distinguished in Figure 3. However, when Au NPs are used to decorate the InGaN NRs, not only is the light absorption of the InGaN NRs in the near-infrared short-wave region increased, but also the absorption region is extended to the near-infrared long-wave region. Enhanced light absorption may be attributed to SPR excitation of Au NPs [24,36,37,38,39,40], and the shift of the long-wavelength absorption edge (1000 nm) is due to the intrinsic absorption characteristics of Au NPs [19]. Interestingly, the sample also shows an absorption peak around 850 nm, which is mainly the absorption peak of InGaN [41]. From Figure 3, it also can be clearly seen that the absorption spectra of different growth concentrations have different absorption intensities. A higher absorption intensity was observed when fabricating the photoelectrode using higher concentration HAuCl_4_ aqueous solutions. This can be ascribed to the higher density of Au NPs decorating the surface of the InGaN NRs, and the larger particle size at higher concentrations with the same conditions [40]. This result is in good agreement with the previous SEM and EDS results.

To study the effect of deposition of Au NPs on the InGaN NRs photoelectrode, InGaN NRs with different densities of Au NPs deposited onto them were prepared. Figure 4a reflects the linear scan curves of different samples under light and dark conditions. From Figure 4a, it can be found that after depositing Au NPs, the photocurrent density of the photoelectrode increases remarkably with the increase of Au NP deposition density. The dark current of the photoelectrode is hardly changed with the deposition of Au NPs, which indicates that the sample has a high crystal quality. However, when the current density of the Au10@NRs reaches its highest value, further improvement of the concentration of HAuCl_4_ aqueous solutions is harmful for device performance, which manifests as a decrease in the photocurrent density as the concentration of HAuCl_4_ aqueous solutions increases. This can be explained by using a higher concentration of HAuCl_4_ aqueous solutions to prepare samples under the same conditions so that the surface of the InGaN NRs is mostly covered by a large amount of Au NPs, which blocks the surface of the InGaN NRs from absorbing light, thereby reducing the generation of photogenerated electron–hole pairs. The high coverage of the Au NPs blocks the contact of the surface of the InGaN NRs with the electrolyte and hinders the water oxidation reaction [26,42]. This shows that the deposition of Au NPs with an appropriate density has a significant improvement in the efficiency of the InGaN NRs PEC water splitting system. In addition, the enhanced photocurrent density can be ascribed to several effects. Firstly, Au NPs increase light absorption by the SPR effect. Secondly, Au NPs capture photogenerated electrons from InGaN NRs for water reduction and reduce the recombination of photogenerated electrons and holes [43,44]. Thirdly, Au NPs generate hot plasmon to excite electrons to directly affect water reduction [45]. Finally, Au NPs form a Schottky contact with the InGaN NR surface, thereby forming a built-in electric field to promote the separation of photogenerated electrons and holes. Figure 4b shows the I–t curves of the bare NRs, Au5@NRs, Au10@NRs, and Au15@NRs under chopped illumination at 0.6 V. All samples showed extremely low current density in the dark. Photocurrent density rapidly increases to its highest value under light conditions. The photocurrent density and on/off ratio of the sample deposited with 10 μM HAuCl_4_ aqueous solution are the largest. The sharp increase in photocurrent density from off to on states indicates that the sample has a significant photoresponse. There is effective generation and separation of photogenerated electrons and holes, and the rapid transport of electrons from the Au NPs to the InGaN NR surface, which promotes the progress of the water splitting. In addition, the photocurrent density of the samples is only slightly reduced after 400 s, which indicates that the samples have extremely strong stability. In summary, samples made with 10 μM HAuCl_4_ aqueous solution show the best PEC water splitting performance.

To further study the enhancement mechanism of the InGaN Au10@NRs, a series of electrochemical measurements was performed and is shown in Figure 5. Figure 5a shows the Nyquist diagram of the sample. This is the use of electrochemical impedance spectra to evaluate the carrier transfer process of the sample, and the radius of the Nyquist plot circle represents the charge transfer resistance. Au10@NRs show smaller charge transfer resistance compared to bare NRs, which indicates that Au NPs can increase the electric conductivity of InGaN NRs, thereby increasing the carrier transport speed between the Au NPs and InGaN NRs interface. Photoelectrode stability is critical for the PEC water splitting system, and Figure 5b shows the photocurrent density–time curve of bare InGaN NRs and InGaN Au10@NRs with an applied bias of 0.6 V in 0.5 M H_2_SO_4_ solution. It is clearly shown that the photocurrent densities of bare InGaN NRs and Au10@NRs have only slightly attenuated after a one-hour test, which indicates that the sample has good stability.

In order to better understand the carrier transport mechanism, the schematic of mechanisms for the enhanced PEC water splitting efficiency of the InGaN NRs@Au photoelectrode is shown in Figure 6. When the photoanode is irradiated under light, the InGaN NRs are excited to generate photogenerated electron–hole pairs. The photogenerated electrons are transferred to the Si substrate through the InGaN NRs, and then to the Pt electrode through an external circuit. Simultaneously, Au NPs on the surface of the InGaN NRs absorb light to generate photogenerated electrons due to the SPR effect. The generated hot electrons are injected into the InGaN conduction band, and electrons accumulate in the InGaN conduction band because of the effect of the Schottky barrier, such that more electrons are transported between InGaN and Si and to the cathode through an external circuit. The synergistic effect of Au NPs reduces the conduction band of InGaN and increases light absorption efficiency. Moreover, due to the Schottky contact between Au and InGaN, a built-in electric field is generated to promote the separation of photogenerated electrons and holes and increase the lifetimes of carriers. More electrons are transferred to the Pt electrode to increase the efficiency of the PEC water splitting system. Therefore, the deposition of Au NPs on the surface of InGaN NRs is a feasible method to improve the efficiency of PEC InGaN water splitting.

## 4. Conclusions

We have successfully prepared the Au NPs@InGaN NRs photoelectrode and controlled the growth density of Au NPs by changing the concentration of the HAuCl_4_ aqueous solution. It can be found that the deposition of Au NPs on the surface of InGaN NRs can significantly improve the efficiency of water splitting. Moreover, the density of Au NPs deposited with 10 μM HAuCl_4_ aqueous solutions is the most suitable. The increasing of photocurrent density is mainly attributed to the SPR effect, and it promotes the separation of photogenerated electron–hole pairs. In addition, the NRs structure of InGaN has a larger body surface area to capture more light, and the directional transfer of the NRs promotes the carrier separation and transfer. In conclusion, this work provides a feasible method for improving the efficiency of semiconductor PEC water splitting and has a promoting effect on the practical application of PEC water splitting.

## Figures and Tables

**Figure 1 nanomaterials-10-00912-f001:**
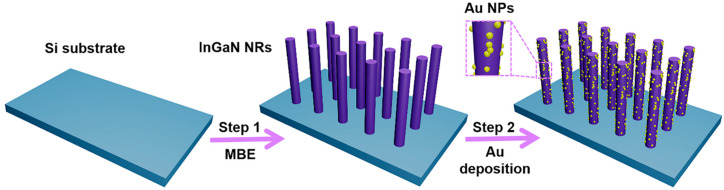
Schematic diagram of the preparation process for decoration of Au nanoparticles (NPs) on InGaN nanorod (NR) arrays.

**Figure 2 nanomaterials-10-00912-f002:**
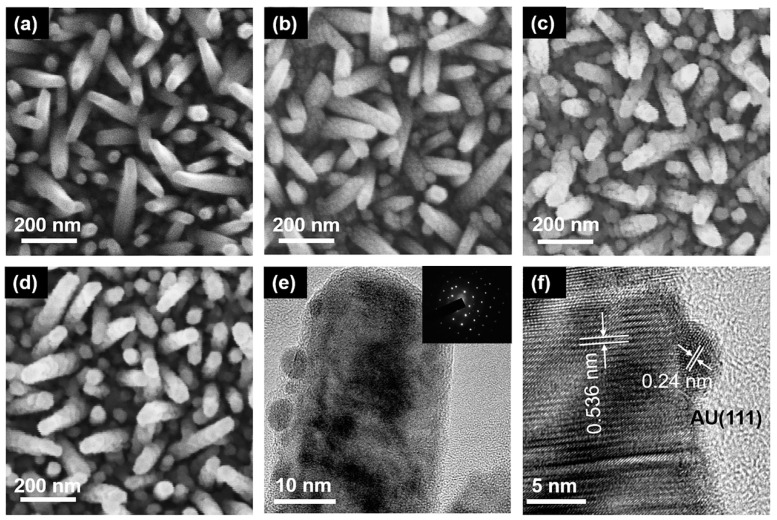
SEM images of InGaN (**a**) bare NRs, (**b**) Au5@NRs, (**c**) Au10@NRs, and (**d**) Au15@NRs. (**e**) TEM image of Au@NR, inset is the corresponding selected area electron diffraction. (**f**) High-resolution TEM image of Au5@NR.

**Figure 3 nanomaterials-10-00912-f003:**
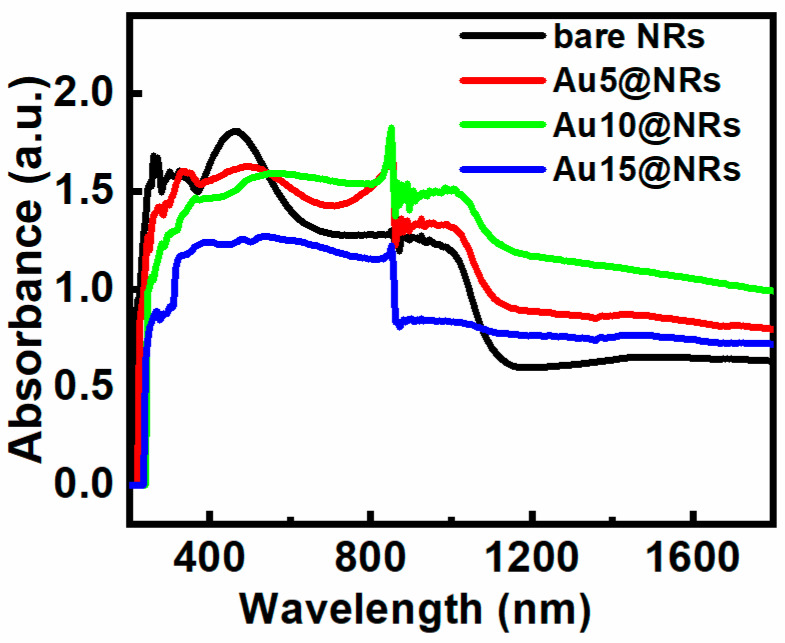
Absorption spectra of bare NRs, Au5@NRs, Au10@NRs, and Au15@NRs.

**Figure 4 nanomaterials-10-00912-f004:**
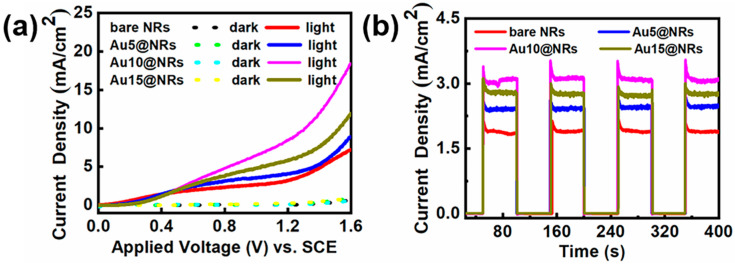
(**a**) Current–potential curves of the bare NRs, Au5@NRs, Au10@NRs, and Au15@NRs photoanodes in 0.5 M H_2_SO_4_ solution in the dark and under 100 mW cm^−2^ irradiation. (**b**) The amperometric I–t curves of the bare NRs, Au5@NRs, Au10@NRs, and Au15@NRs under chopped illumination at 0.6 V vs. reversible hydrogen electrode (RHE).

**Figure 5 nanomaterials-10-00912-f005:**
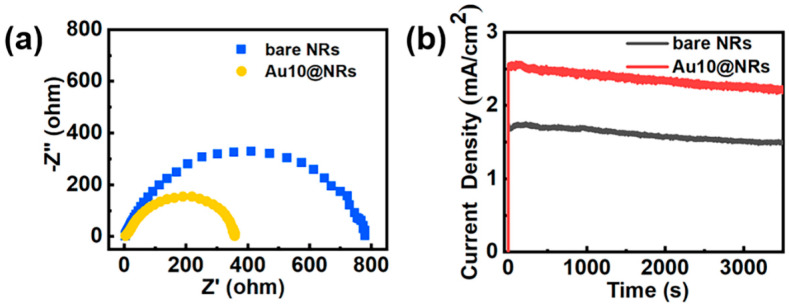
(**a**) Nyquist plots of bare NRs and Au10@NRs. (**b**) Stability of the bare NRs and Au10@NRs photoanodes for photoelectrochemical (PEC) the oxygen evolution reaction (OER).

**Figure 6 nanomaterials-10-00912-f006:**
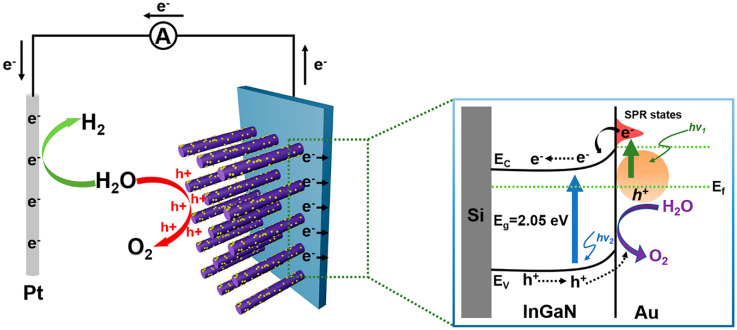
Schematic of mechanisms for understanding the enhanced PEC water splitting efficiency of the Au@InGaN NR photoelectrode.

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
