# Peer review of "InGaN Nanorods Decorated with Au Nanoparticles for Enhanced Water Splitting Based on Surface Plasmon Resonance Effects"

_nanomaterials, 2020, doi:10.3390/nano10050912_

Round 1

Reviewer 1 Report

The authors present a report studying the photosensitization of InGaN NRs with small Au nanoparticles, for the purpose of photoelectrochemical water splitting. Although this material has been extensively studied in this context, the number of works studying plasmonic-enhanced strategies using InGaN are still limited, so exploring them is probably useful.

However, this report does that in a very unclear manner, with inconclusive results and important conceptual mistakes in the discussion of the plasmonic effects. I do not recommend its publication. Below I discuss some of the problems that I find with the manuscript:

  • The introduction does not discuss relevant existing literature on plasmonic-enhanced InGaN systems for water splitting. A cursory search reveals, for instance (Yaonan Hou 2019 of Photonics for Energy 9, 026001, DOI: 10.1117/1.JPE.9.026001) and (Yimeng Sang et al 2020 Semicond. Sci. Technol. 35 025017, DOI: 10.1088/1361-6641/ab615a)
  • The discussion on plasmonic mechanism for the enhancement of the water splitting process is too general, and with clear mistakes. For instance, the authors claim that scattering is one of the main mechanisms contributing to the enhancement, although this is not only not supported by the evidence but also highly unlikely because 5 nm Au nanoparticles will scatter virtually no light.
  • The optical spectra of the samples (Fig. 3) is very confusing, and does not support the short discussion around it. What is the source of absorption for wavelengths above the InGaN and Si band gaps? Why does the gold amplify across the whole spectrum, when the resonance of 5 nm Au NPs will occur at around 520 nm? What are the strange signals at around 850 nm? What is the resonant-looking peak at around 500 nm for the bare NRs? Overall, this figure shows that, consistently with the later data in the paper, a larger absorbance is seen for the Au10@NRs sample, but without clarifying at all why this is the case.

Reviewer 2 Report

This article present the decoration of IbGAN nanorod substrates with AuNPS, and their use for photoelectrochemical water splitting.

The decorated samples are well characterized and provide a noticeable improvement over the untreated one. This article meets the criteria of quality for a publication in Nanomaterials, pending some minor points to solve,

1) The light absorption analysis is a bit unclear. First, there is a noticeable shift for Au15 around 800 nm, likely due to a mis calibration of the instrument during the lamp change. These data should be acquired again if possible. Second, the bare sample exhibits a noticeable band around 500 nm, that progressively disappear when gold is deposited. What is the origin of this band?

2) Where the sample metallized before the SEM observation?

3) How was collected the nanorod for TEM observation?

4) p5 l150, Figure 3 is called figure 5.

5) In the legend of figure 3, the gold loading should be added for the TEM sample.

Reviewer 3 Report

Authors:  Qing Liu, Jiang Shi, Zhenzhu Xu, Bolin Zhang, Hongliang Liu, Yinlei Lin, Fangliang Gao, Shuti Li and Guoqiang Li

Title of the manuscript:

InGaN Nanorods Decorated with Au Nanoparticles for Enhanced Water Splitting Based on Surface Plasmon Resonance Effects

The authors successfully prepared InGaN nanorods (NR) on Si substrate and decorated with Au nanoparticles (NP) with various areal densities. These nanocomposite structures are suitable for solar water splitting (=> hydrogen production) as pointed out by the authors. Using the Au NP decorated semiconductors for enhanced solar water splitting in connection with the surface plasmon resonance is already mentioned in review paper of Zhang et al. (Zhang, H., Chen, G., & Bahnemann, D. W. (2009). Photoelectrocatalytic materials for environmental applications. Journal of Materials Chemistry, 19(29), 5089-5121.). This type of structures (Semiconductor+Au/Ag NP) and the role of the surface plasmon resonance were also studied by Zhou et al. (Zhou, X., Liu, G., Yu, J., & Fan, W. (2012). Surface plasmon resonance-mediated photocatalysis by noble metal-based composites under visible light. Journal of Materials Chemistry, 22(40), 21337-21354.). Therefore, I recommend that the authors cite also the above mentioned papers, adding them as references to refs. [15,16]. The present manuscript is useful since nobody has studied photocatalysis of InGaN NR with Au NP before. However, there are papers published on solar water splitting where authors used similar materials: E.g. Kibria, M. G., Nguyen, H. P., Cui, K., Zhao, S., Liu, D., Guo, H., ... & Mi, Z. (2013). One-step overall water splitting under visible light using multiband InGaN/GaN nanowire heterostructures. ACS nano, 7(9), 7886-7893.
Chowdhury, F. A., Mi, Z., Kibria, M. G., & Trudeau, M. L. (2015). Group III-nitride nanowire structures for photocatalytic hydrogen evolution under visible light irradiation. APL Materials, 3(10), 104408.

In abstract, the authors mention the need to increase the photo-conversion efficiency: “But what stands in the way for the practical application of this technology is the low conversion efficiency, which can be promoted by optimizing the material structure and device design, surface functionalizing.” It would be useful to obtain photo-conversion efficiency for the presently studied material and compare it with photo-conversion efficiencies of some other solar water splitting materials (optional).

Other suggestions follow:

  1. I am not sure whether the convention of putting references behind a sentence is right (…sentence. [x]).
  2. Page 2 Line 85 …dried with instead of “dry with”
  3. Page 2 Line 93 …were…blown dry with… instead of “blow dry with”
  4. Page 4 Line 124 …which indicates the high quality…
  5. Page 4 Line 134 the well-resolved highly crystalline structure…
  6. Page 4 Line 137 S1: In which order the images (a), (b), (c) correspond to Au5@NRs, Au10@NRs, Au15@NRs? What are the Au NP sizes for each concentration? Diameter 5 nm (Line 136) is for which group of Au5@NRs, Au10@NRs, Au15@NRs?
  7. Page 5 Line 145 whereas instead of “…while…”
  8. Page 5 Lines 145 and 148 For readers, it may be helpful to specify the near-infrared short-wave and near-infrared long-wave regions, e.g. (780 nm – 1100 nm)…
  9. Page 5 Line 150: Please, specify the wavelength of the mentioned absorption edge in the text.
  10. Page 5 Line 150: Figure 3 instead of Figure 5.
  11. Page 5 Line 154: It seems that the authors did not show, in the previous text, that they got larger Au particles for higher concentrations. In their text, there is only one particle size mentioned (~5 nm – line 136)
  12. Page 5 Lines 164-166. The sentence “However, when …” is not clear.
  13. Figure 4: The authors may consider using the same colors in Figs a and b as for denoting concentrations. For example, in Fig. 4a, blue color corresponds to Au5@NRs but in Fig. 4b, blue color corresponds to Au15@NRs.
  14. Caption of Figure 6 and Line 213: I am not sure if term “water splitting slitting efficiency” is right or perhaps just only “water slitting efficiency”?
  15. Page 7 Line 224 and Page 8 Line 232. It is not a formal style to start sentence with “And”.

Round 2

Reviewer 1 Report

Although I appreciate the effort by the authors in improving their manuscript, the changes have not improved the main problems in the interpretation of the optical data, and the new text (although corrects some problems) adds new mistakes. The small Au nanoparticles that the authors have used (below 20 nm) will resonate below 600 nm, not at ~850 nm, and the rest of the explanation regarding the optical profile of the sample is still imprecise and largely incorrect.

I recommend the rejection of the revised manuscript.
